# Analysis of Spatiotemporal Changes in the Gravitational Structure of Urban Agglomerations in Northern and Southern Xinjiang Based on a Gravitational Model

**Difan Liu** [1,2], **Yuejian Wang** [1,2,*], **Lei Wang** [1,2], **Liping Xu** [1,2], **Huanhuan Chen** [1,2] and **Yuxiang Ma** [1,2]

1  College of Science, Shihezi University, Shihezi 832000, China; liudifan@stu.shzu.edu.cn (D.L.); wanglei@dlut.edu.cn (L.W.); xlpalw@163.com (L.X.); chenhuan@shzu.edu.cn (H.C.); mayxiang@stu.shzu.edu.cn (Y.M.)
2  Key Laboratory of Oasis Town and Mountain Basin System Ecology Corps, Shihezi 832000, China
*  Correspondence: wangyuejian0808@163.com; Tel.: +86-1809-993-9983

**Abstract:** The urban agglomeration plays a significant role in enhancing integrated regional development. Nevertheless, the expansion of urban agglomerations has demonstrated a lackluster ability to attract cities. Presently, finding solutions to stabilize the existing urban strength and effectively extend attraction to neighboring cities has become a crucial matter. This study adopts the enhanced comprehensive attraction model, fracture point model, and radiation radius model to examine the level of city attraction, intensity of radiation, and range of radiation in the northern and southern Xinjiang city clusters between 2010 and 2020. Based on the analysis, the following conclusions are drawn: (1) the comprehensive strength and city attraction of cities in the northern Xinjiang region is higher than that of the southern Xinjiang region; (2) the intensity of spatial connection between cities in the northern and southern Xinjiang regions is gradually increasing, and the intensity of spatial connection of cities in the northern Xinjiang region is significantly greater than that in the southern Xinjiang region. The intensity of spatial connection between cities in the northern and southern Xinjiang regions is significantly greater than that in the southern Xinjiang region; (3) the central role of the central cities in the northern and southern Xinjiang regions is weakening, the development of cities in the region is gradually becoming unified and coordinated, and regional integration is gradually being strengthened. This study reveals the similarities and differences in urban development in the north and south of Xinjiang and provides important theoretical reference value for regional urban development.

**Keywords:** urban gravity; urban linkage strength; gravity modeling; urban agglomerations; north–south border

## 1. Introduction

Urban development in China has accelerated due to the growth of the social economy and the reform and opening up policies that have framed the country's policies. From 1978 to 2022, the urban population of China increased from 172.45 million to 920.71 million in 2022, an increase of 74.826 million people, which exceeds the total population of Europe. The urbanization rate in China has increased from 19.39% to 65.22% [1], and now it still maintains a high-speed growth. Meanwhile, with the accelerated development of industrialization and urbanization in China, urban spatial expansion and the expansion of urban influence have led to a large-scale emergence of urban agglomeration, making urban agglomerations the main form of urbanization. An urban agglomeration is a group of cities within a specific geographic area that have a significant variety of attributes, levels, and types [2]. Urbanization is mostly driven by one or more megacities or major cities acting as the hubs. These cities depend on the environment, transportation and information networks, and internal and external connections with the surrounding areas [3–5].

China's development is positively impacted by urban agglomerations as well [6,7]. Urban agglomerations were considered by China as a crucial tool for advancing the development of new urbanization during the 11th Five-Year Plan period. The nation established 19 significant urban agglomerations during the 13th Five-Year Plan period, including the Yangtze River Delta urban agglomeration, northern Xinjiang, and the urban agglomerations of Beijing, Tianjin, and Hebei. The 14th Five-Year Plan and the 2035 long-term goals subsequently further optimized the regional definition of urban agglomerations. In order to encourage regional integration of urban agglomerations, a new national urbanization development strategy that is built on agglomerations and prioritizes human development has been proposed.

In recent years, China's economy and society have made rapid progress, and regional integration has been developing rapidly, affecting the spatial connections between cities and within and outside clusters. The economic coordination and reasonable spatial layout of urban clusters have become important factors of competitiveness [8–10]. To this end, scholars have conducted research on the regional economy and spatial connections of urban agglomerations, focusing on the strength and potential of connections, the structure and evolution of connections, and the optimization of radiation patterns. The perspectives include provinces, metropolitan areas, and urban agglomerations. The research methods include social network analysis, urban factor flow, and gravity model [11,12], among which the gravity model is the most widely used [13–15]. Representing the size of regional economic connections or spatial interactions through gravity is one of the important methods used for studying urban spatial connections. Zipf was the first to apply the law of universal gravitation in physics to the study of interactions between cities [16]. Early urban gravity models were used to study the spatial structure of cities and later gradually introduced into research on regional economic connections, urban interaction structures, and spatial structures of urban connections.

When reviewing previous studies, there is a broad spectrum of findings about the geographical relationships between urban agglomerations. Relatively little research has been conducted on the regional linkages of cities in impoverished areas; instead, the majority of studies on the gravity of urban agglomerations in China concentrate on developed urban agglomerations like the Yangtze River Delta and the Beijing–Tianjin–Hebei region [17–19]. Xinjiang, which connects eight Eurasian nations outside and five northwest provinces internally, has become the center of opening up to the West with the ongoing development of the "Belt and Road" plan [20,21]. As the hub of the Silk Road Economic Belt, Xinjiang has become more and more ideal in recent years due to its rapid infrastructural development and increased degree of international economic and trade interaction. Xinjiang has made it plain that it will intensify the new urbanization strategy's implementation during the 14th Five Year Plan period. It will also develop the Urumqi metropolitan area, construct the northern Xinjiang urban belt, establish the southern Xinjiang urban agglomeration, and form a moderately sized, fully operational, and rationally arranged urban system. Simultaneously, Xinjiang's urbanization and regional integration have been propelled by the existence of urban agglomerations on the northern slope of the Tianshan Mountains, as well as oasis urban agglomerations like Kashgar and Korla. Even with its 2022 urbanization rate of 57.89%, Xinjiang remains less urbanized than the country as a whole. In addition, there are disparities in the degree of urbanization growth between northern and southern Xinjiang because of variations in their respective natural and economic bases [22]. Southern Xinjiang's economic development is far behind that of northern Xinjiang because of the region's harsh environment and inadequate transportation systems. Thus, examining the spatial relationships between the cities in the Xinjiang region and elucidating the features and issues of the urban agglomeration's development are crucial for both the theoretical and practical aspects of the agglomeration's creation and growth.

This study selected the urban agglomeration in the Xinjiang region as the research object, with nine cities selected in the northern Xinjiang region and seven cities selected in the southern Xinjiang region. We used the improved comprehensive gravity model, fault

zone model, and GIS to analyze the spatial connectivity characteristics and urban gravity range of cities in the urban agglomeration of northern and southern Xinjiang from 2010 to 2020. The problem we need to solve is as follows: 1. Explore the spatial interaction characteristics of urban agglomerations in northern and southern Xinjiang. 2. Explore the differences in urban patterns between northern and southern Xinjiang. 3. Based on the layout characteristics of urban agglomerations in northern and southern Xinjiang, propose appropriate urban development suggestions.

## 2. Study Area and Data Sources

### 2.1. Study Area Overview

Xinjiang is located in the west of China, with a geographical position between 73°40′~96°18′ E and 34°25′~48°10′ N. It is the center of Asia and Europe, with a vast territory, accounting for one-sixth of the land area of China. Xinjiang is also an important area along the Silk Road and an important transportation hub, and it has abundant mineral resources and a special topography of "three mountains and two basins", which divides Xinjiang into southern and northern Xinjiang by the Tianshan Mountains. However, due to the limitations of Xinjiang's geographical location, resources, and development base, Xinjiang's urban development is slow, and the level of urbanization is low. As of 2020, the urbanization rate of Xinjiang reached 50.91%. However, there are obvious geographical differences in the development of urbanization in Xinjiang, with cities in northern Xinjiang (Urumqi, Karamay, Shihezi, Changji, Fukang, Yining, Kuitun, Tacheng, and Altay) developing more rapidly and having higher urbanization levels, while cities in southern Xinjiang (Kullu, Aksu, Atushi, Kashgar, Hotan, and Alar) and the cities in southern Xinjiang (Kulle, Aksu, Atushi, Kashgar, Hotan, Alar, Tumushuk) have a poor economic base, lack of resources, and limited transportation, making the urbanization level in southern Xinjiang lower.

### 2.2. Data Source and Processing

The data for this study mainly includes socio-economic data, the shortest distance between cities, and road traffic market data from a total of 16 cities in northern and southern Xinjiang. Among them, the socio-economic data are sourced from the 2010–2010 China Urban Statistical Yearbook, Xinjiang Urban Statistical Yearbook, Xinjiang Urban National Economy and Social Development Statistical Yearbook, Xinjiang Uygur Autonomous Region Urban County Construction Statistical Yearbook, Xinjiang Statistical Yearbook, and survey data from related websites. The data on the shortest distance between cities, also known as road travel time, is obtained by searching for the shortest transportation distance between two city centers on Baidu Maps. The raw data were standardized using SPSS 26 data processing software, and the data processing results were imported into ArcGIS 10.3 for spatial network analysis and visualized.

## 3. Research Methodology

### 3.1. Construction of a Comprehensive Gravity Model

The physics model of gravity is used to represent the idea that interactions between spatial objects become less significant as distance increases [23]. Urban research has adopted this approach; however, there are certain restrictions [24]. Therefore, this work improves the conventional gravity model in the following ways by fusing real-world applications with the findings of previous research.

First off, population or GDP size alone cannot adequately capture the influence of cities [25], so they cannot be used as a sole indicator of urban quality. Cities create demand that is complementary to one another and draw people in due to their overall strengths in a variety of areas, including infrastructure, society, and the economy. Therefore, in order to quantify urban "quality" effectively, we need an indicator that is more thorough. In order to create a thorough method for evaluating urban strength, this study took into account the elements that influence urban disparities and chose 11 evaluation factors from 4 categories: economic scale, land scale, social development level, and ecological

environment preservation. The number of people living in cities, GDP per capita, the share of secondary and tertiary industries, and the average salary of employed people all serve as indicators of the economic scale. The built-up area, population density, and per capita urban road area all serve as indicators of the urban land scale. We can utilize the popularity of urban gas and retail sales of consumer goods to gauge the degree of urban social development and the rate of built-up area green coverage and per capita public green space area to gauge the amount of ecological environment protection. To describe the "quality" of the gravity model city, we can compute the comprehensive strength of a city using the evaluation factors mentioned above. See Table 1.

**Table 1.** Evaluation index system for urban comprehensive strength.

| Level 1 Indicators | Secondary Indicators | Indicator Weights |
|---|---|---|
| Size of economy | Urban population (10,000) | 13.968 |
| | GDP per capita (persons/USD) | 7.931 |
| | Share of secondary and tertiary industries (%) | 2.123 |
| | Average wage of employed workers (dollars) | 9.627 |
| Landholding | Built-up area (Km$^2$) | 21.444 |
| | Population density (persons/Km$^2$) | 3.539 |
| | Urban road area per capita (m$^2$) | 5.459 |
| Level of social development | Total retail sales of social consumer goods (million USD) | 26.499 |
| | City gas penetration rate (%) | 1.218 |
| Eco-Environmental Protection | Public green space per capita (m$^2$) | 6.325 |
| | Greening coverage of built-up areas (%) | 1.867 |

A composite scoring method was utilized to rate the overall strength of the cities in the northern and southern border regions:

(1) Data standardization. In this paper, we choose the method of standardization of extreme deviation to standardize the data of the indicators of each city in order to eliminate the influence of the different data outline, which is publicly announced as follows:

$$M_{ij} = \left(X_{ij} - minX_j\right) / \left(maxX_j - minX_j\right) \tag{1}$$

where $M_{ij}$ is the normalized value of the $i$th project under the $j$th indicator, $X_{ij}$ is the indicator value of the $i$th city under the $j$th indicator, and $minX_j$ and $maxX_j$ are the maximum and minimum values of the $j$th indicator, respectively.

$$I_{ij} = M_{ij} / \sum_{i=1}^{m} M_{ij} + 0.001 \tag{2}$$

where $I_{ij}$ is the normalized value, $m$ is the number of cities, and 0.001 is an overall shift to the right to prevent the presence of a value of 0 and facilitate subsequent calculations.

(2) Entropy weight method to determine the weight of indicators. We can calculate the entropy value and entropy weight of each indicator $e_j$, $W_j$; the formula is as follows:

$$e_j = -1/\ln m \sum_{i=1}^{m} I_{ij} \ln I_{ij} \tag{3}$$

$$W_j = 1 - e_j / \sum_{j=1}^{n} \left(1 - e_j\right) \tag{4}$$

where, $e_j$ ($0 \leq e_j \leq 1$) is the entropy value of the $j$th indicator, $-1/\mathrm{Inm}$ is the information entropy coefficient, $W_j$ is the entropy weight value of the $j$th indicator, and n is the number of urbanization indicators.

(3) Composite score. We can calculate the score of the urbanization level of the city to reflect the level of development of the city by the size of the value. The formula is

$$S_{ij} = \sum_{i=1}^{n} I_{ij}W_j \tag{5}$$

where $S_{ij}$ is the value of each indicator for the $i$th city, $I_{ij}$ is the normalized value of the urban development level indicator, and $W_j$ is the weight of each urbanization indicator.

Second, the distance-attenuation effect between cities is not well represented by the direct distance in conventional gravity models. Cities are linked together by a variety of networks, and advancements in communication and transportation technologies have alienated the idea of distance [26,27]. Consequently, a more precise measurement of the "distance" between cities is required, taking into account all relevant factors, including time, cost, and route. In this article, the direct distance variable included in conventional gravity models is replaced by time distance. A lot of the cities in Xinjiang are not easily accessible by air or rail because of the disparities in transportation convenience between the region's north and south. Consequently, after careful analysis, the distance between cities in this article is calculated using the intercity highway's quickest travel time.

Thirdly, as regional collaborative development continues to grow, cities strive for greater levels of development within their borders to compensate for shortcomings or capitalize on advantages [28]. They also frequently go outside their borders for external complementarity and to meet their demands from a variety of angles. At this point, cities begin to create regional relationships that fully capture their radiation capability [29]. There must be variations in each city's appeal or an asymmetry in the attractiveness across cities as a result of their varying levels of development [30]. As a result, cities are drawn to one another in both directions [31–33]. The degree to which a city attracts other nearby cities can be represented by its positive attraction, and the degree to which it accepts radiation from other nearby cities can be represented by its negative attraction. This study uses the complete strength of two cities to increase the gravitational constant G based on previous research:

$$K_a = M_a / (M_a + M_b) \tag{6}$$

In the formula, $K_a$ is the ratio of the comprehensive strength of city $a$ to the sum of the comprehensive strength of the two cities, representing the direction of the spatial connection between city a and city b. The closer the value of $K_a$ is to 1, the more it indicates that the city connection is positively radiating to the outside world, and the closer it is to 0, the more it indicates that the city connection is negatively attracted to the outside world, and the two situations indicate that there is a demand between the cities; when the value of $K_a$ is close to 0.5, it indicates that the two cities are complementary to each other, and the connection is relatively balanced.

Ultimately, the improved gravitational model for this study is as follows:

$$F_{ab} = K_a \frac{M_a M_b}{D_{ab}} \tag{7}$$

Improving the gravitational constant using the magnitude of the combined strength between the two cities can adequately express the asymmetry of the attraction between the cities, so the total intensity of external ties and the intensity of accepting external ties, i.e., the positive and negative attraction, of the city a can also be calculated:

$$F_{ab} = K_a \frac{M_a M_b}{D_{ab}}, F_{ba} = K_b \frac{M_a M_b}{D_{ab}} \tag{8}$$

$$P_a = \sum F_{ab}, \ N_a = \sum F_{ba} \tag{9}$$

In Equation (8), $F_{ab}$ is the spatial linkage intensity from city a to city b, which is regarded as the positive attraction of city a, while $F_b a$ is the spatial linkage intensity from

city b to city a, which is regarded as the negative attraction of city a. In Equation (9), Pa is the sum of linkage intensity between city a and all other cities, indicating the overall degree of the city's external radiation-driven intensity; Na is the sum of the intensity of radiation-driven intensity of city a by all other cities, which is also known as the sum of the intensity of the dependence on the attraction of other cities.

*3.2. Urban Attraction Model*

(1) Fracture point formula

Convis proposes the Breaking Point method to delineate the range of attractiveness of a city, theorizing that the range of attractiveness of a city depends on the level of the city's overall strengths as well as the distance between the cities [34,35]. According to the theory, a city's attractiveness range depends on its level of comprehensive strength and the distance between the cities [36]. The more comprehensively strong a city is, the wider its gravitational attractiveness spectrum can radiate, but as the distance between the two cities grows, the attractiveness of the city gradually weakens until it reaches zero, at which point it becomes insignificant. Based on the foregoing, this study interprets the inter-city distance as the highway distance between cities, calculates the city's comprehensive strength score as a proxy for its size, and modifies the fracture point calculation as follows:

$$D_a = D_{ab} / \left( 1 + \sqrt{M_b / M_a} \right) \tag{10}$$

where $D_a$ is the distance from City a to the rupture point; $D_{ab}$ is the highway distance between City a and City b; and Ma, Mb are the values of the urban development level of Cities a and b.

(2) Field strength formula

$$F_{aP} = M_a / D_{aP}^2 \tag{11}$$

where point $P$ is the fracture point, $F_{aP}$ is the magnitude of gravitational force of City a at the fracture point $P$, $M_a$ is the city size of City a, and $D_{aP}$ is the distance from City a to the fracture point P.

(3) Gravitational radius formula [37,38]

The radius of attraction of the city can reflect the range of radiant energy of the city and can more intuitively reflect the attraction situation between cities; the formula is:

$$R = \sqrt{M_a / F} \tag{12}$$

where $R$ is the radius of the city's radiation, $M_a$ is the combined strength of City a, and $F$ is the strength of the city's boundary field.

**4. Results and Analysis**

*4.1. Spatial and Temporal Characterization of Urban Gravity*

The gravity values of the cities in northern and southern Xinjiang may be computed using the revised gravity model, which can indicate changes in urban development level and spatial connection intensity of cities in various phases from 2010 to 2020.

It is clear from the results of Tables 2 and 3 that the comprehensive strength and comprehensive city attractions of cities in the northern Xinjiang region are generally higher than those of cities in the southern Xinjiang region. The comprehensive city attraction and the quality of the cities in northern and southern Xinjiang both showed a steady growth trend from 2010 to 2020, but the four cities in northern Xinjiang—Shihezi, Changji, Fukang, and Yining—saw a slight decline in both of these metrics from 2016 to 2020. According to the gravity model, a city's attractiveness is primarily influenced by its level of development and its proximity to other cities. The quality of cities and the strength of inter-city ties in Xinjiang are characterized by "high in the north and low in the south" due to the comparative strength of northern border cities being higher than southern border cities and

the distance between southern border cities being much greater than that between northern border cities.

**Table 2.** Urban comprehensive strength and comprehensive gravity of sorthern Xinjiang from 2010 to 2020.

| Age | 2010 | | 2013 | | 2016 | | 2020 | |
|---|---|---|---|---|---|---|---|---|
| F | Urban Comprehensive Strength | Urban Comprehensive Gravity | Urban Comprehensive Strength | Urban Comprehensive Gravity | Urban Comprehensive Strength | Urban Comprehensive Gravity | Urban Comprehensive Strength | Urban Comprehensive Gravity |
| Urumqi | 0.5199 | 2380.8623 | 0.7149 | 4706.5183 | 0.7837 | 6149.6849 | 0.7862 | 6197.707 |
| Karamay | 0.214 | 419.2699 | 0.3262 | 908.0753 | 0.3443 | 1101.5086 | 0.3737 | 1253.0383 |
| Shihezi city | 0.159 | 370.8971 | 0.2243 | 737.7493 | 0.2515 | 957.7472 | 0.2376 | 887.7689 |
| Changji city | 0.1308 | 368.5206 | 0.1928 | 791.7402 | 0.2466 | 1257.7024 | 0.2375 | 1183.108 |
| Fukang city | 0.0953 | 186.6684 | 0.1735 | 562.7752 | 0.231 | 957.4028 | 0.2106 | 825.5427 |
| Yining city | 0.143 | 106.2632 | 0.1773 | 171.721 | 0.2172 | 256.3679 | 0.1939 | 216.9976 |
| Kuitun city | 0.1363 | 388.9559 | 0.1764 | 672.3875 | 0.1942 | 836.1194 | 0.227 | 1076.102 |
| Tacheng city | 0.0859 | 68.6727 | 0.1167 | 127.0178 | 0.1571 | 221.2435 | 0.1693 | 251.7137 |
| Altay city | 0.1088 | 87.6592 | 0.0973 | 82.5996 | 0.1568 | 194.593 | 0.1655 | 213.6615 |

**Table 3.** Urban comprehensive strength and comprehensive gravity of southern Xinjiang from 2010 to 2020.

| Age | 2010 | | 2013 | | 2016 | | 2020 | |
|---|---|---|---|---|---|---|---|---|
| F | Urban Comprehensive Strength | Urban Comprehensive Gravity | Urban Comprehensive Strength | Urban Comprehensive Gravity | Urban Comprehensive Strength | Urban Comprehensive Gravity | Urban Comprehensive Strength | Urban Comprehensive Gravity |
| Korla | 0.1954 | 87.2826 | 0.2627 | 155.0794 | 0.2821 | 199.5615 | 0.3195 | 261.3781 |
| Aksu | 0.125 | 106.6557 | 0.1839 | 210.3098 | 0.1956 | 272.8875 | 0.2296 | 373.9949 |
| Artux | 0.0671 | 57.0835 | 0.0939 | 108.7569 | 0.1275 | 195.7921 | 0.1616 | 298.635 |
| Kashgar city | 0.1467 | 145.7485 | 0.1949 | 258.3227 | 0.2385 | 406.9163 | 0.2533 | 498.4918 |
| Hotan city | 0.0909 | 37.6211 | 0.1256 | 69.7633 | 0.1212 | 74.1743 | 0.1607 | 123.2423 |
| Alaer | 0.0924 | 62.6546 | 0.1127 | 97.1373 | 0.1503 | 166.4638 | 0.1803 | 236.634 |
| Tumxuk | 0.104 | 75.3698 | 0.1118 | 97.0606 | 0.2135 | 281.3145 | 0.2287 | 340.7354 |

We used Origin 2021 software's Circos chord diagrams to visualize the outcomes. In its original form, the chord diagram was meant to depict the intricate connections between genes. We visualized the results using Circos chord diagrams in Origin software [39]. This article mainly displays the spatial connection strength between multiple cities through a proportional chord diagram. Circles are divided into different arcs according to the number of cities, and the length of the arcs represents the proportion of the city's gravity. Taking the 2010 North Xinjiang City Chord Diagram as an example, Urumqi's urban gravity accounts for 54% of the city's gravity in the North Xinjiang region, and the connecting lines between different arcs represent the direction of the city's gravity. The color of the connecting lines indicates that the city is attracted more—for example, Urumqi. All connecting lines are in the color of other cities, indicating that Urumqi is in a state of attraction to other cities. The thickness of the connecting lines between cities indicates the strength of mutual attraction between cities. For example, the connecting line between Urumqi and Changji is the thickest, with the strongest attraction between cities.

Table 2 demonstrates that from 2010 to 2020, Urumqi's cities had substantially higher comprehensive strength and gravity than other cities in northern Xinjiang, consistently ranking top and emerging as the only stable "central city" in the area. Figure 1 shows that Changji, Fukang, Shihezi, and Kuitun are more prevalent in Urumqi than Karamay, Yining, Tacheng, and Altay. The comprehensive strength of the cities of Changji, Kuitun, and Yining has not changed considerably over the years, but their comprehensive attraction has. When their position is examined, it is discovered that Changji is close to Urumqi. Kuitun is situated in the middle of the northern Xinjiang metropolis, a reasonable distance from other cities, although it is some distance from Urumqi. Because of its distance from Urumqi and other cities and its location in the westernmost section of the region, Yining has a lower comprehensive attraction and a greater comprehensive strength than other nearby cities.

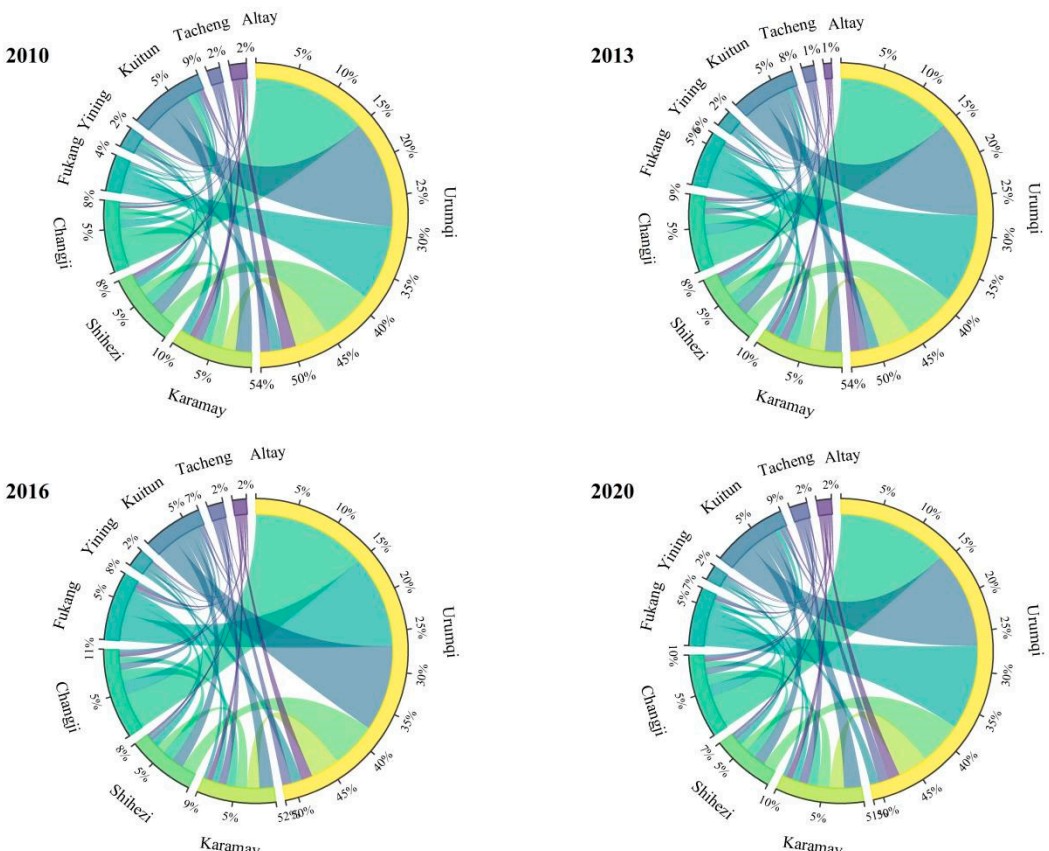

**Figure 1.** Urban connection strength in northern Xinjiang from 2010 to 2020.

Table 3 demonstrates that from 2010 to 2020, the overall strength of Kashgar and Korla cities was comparatively outstanding. In terms of overall strength, Tumxuk surpassed Aksu in 2016 and 2020, moving up to third place, while Aksu came in third in 2010 and 2013. Figure 2 shows that while Kashgar and Aksu have the most pronounced urban spatial connections, Artux, Alaer, and Hotan have continued to have reduced spatial ties. Urban comprehensive strength is increasing in Tumxuk, and a distinctive characteristic is that Korla has the highest urban comprehensive strength in southern Xinjiang. However, because Korla is remote from other cities and has higher comprehensive strength and lower comprehensive attraction, its urban spatial connection strength is only average.

### 4.2. Characterization of Urban Spatial and Temporal Linkages

A city's *p* value represents its total economic radiation from the outside world, while its *n* value represents the total economic radiation that the city depends on from other cities. A city's *p* and *n* values show how close it is to its adjacent cities, as well as the interconnections and power of the inter-city radiation. The *p* value and *n* value of the cities in the north and south from 2010 to 2020 are calculated using Equations (8) and (9), and the structure of the urban connectivity is further examined.

The positive and negative spatial relationships between cities in the northern Xinjiang region were clearly stronger than those in the southern Xinjiang region from 2010 to 2020, as can be seen in Figures 3 and 4. Since 2016, the rate of growth of urban spatial connections has decreased, and the intensity of urban spatial linkages in northern and southern Xinjiang has progressively increased.

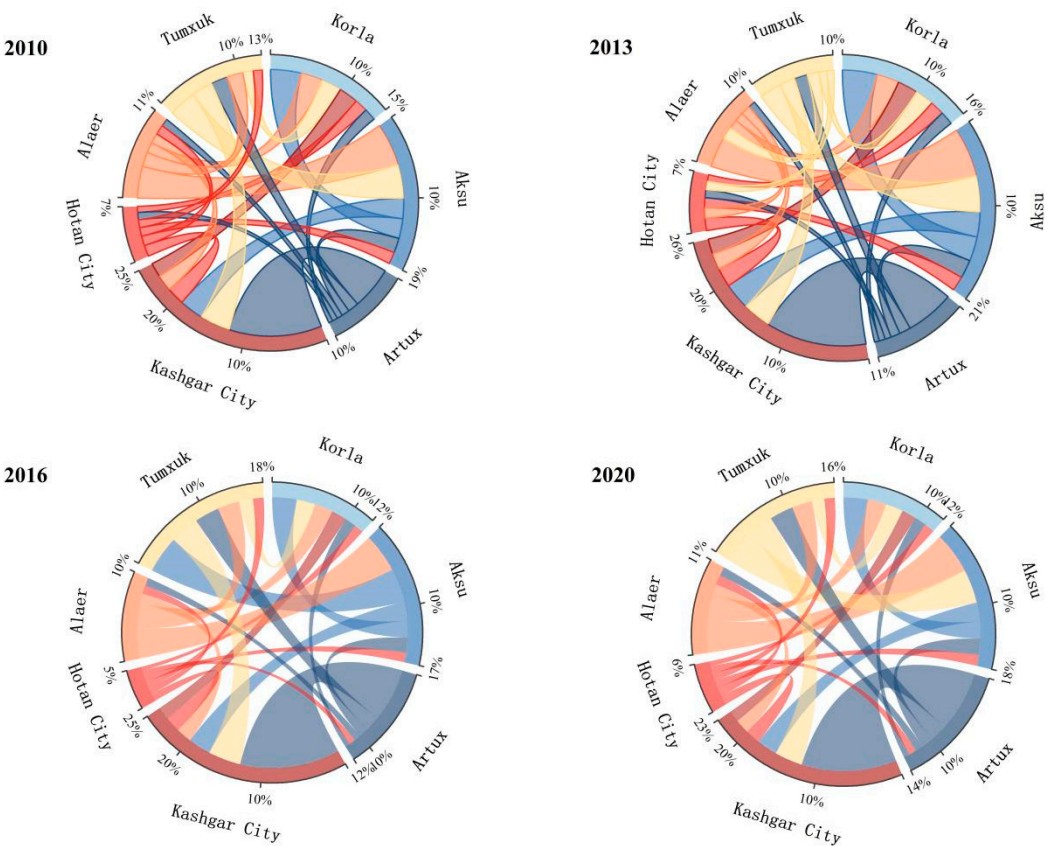

**Figure 2.** Gravity values of cities in southern Xinjiang from 2010 to 2020.

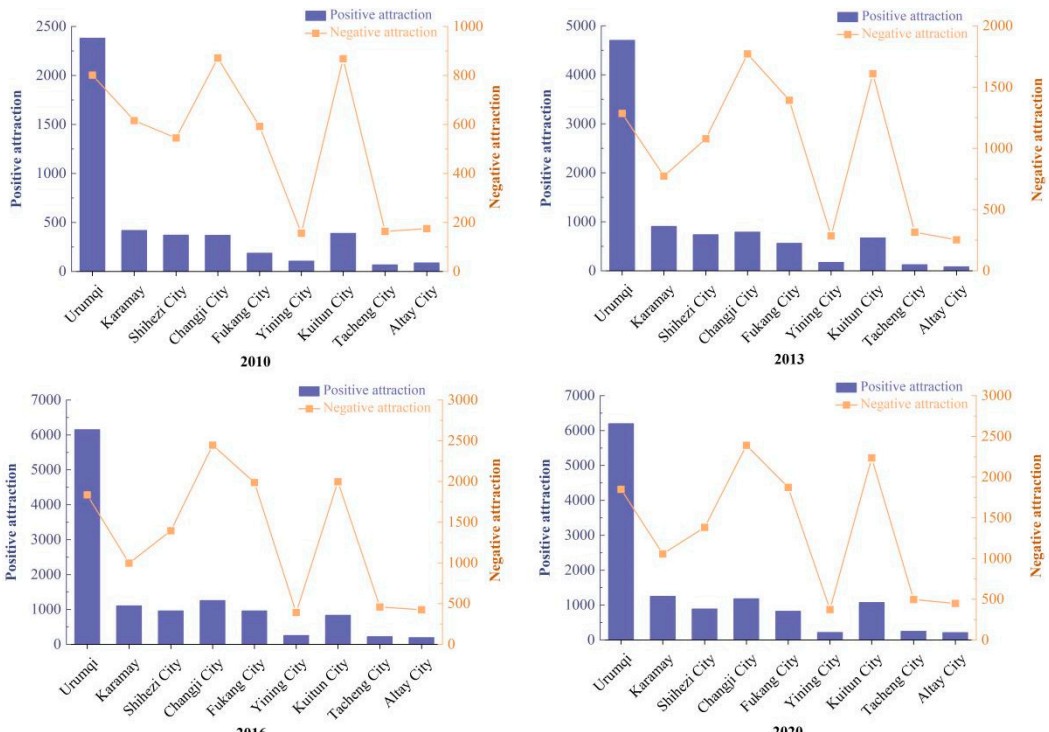

**Figure 3.** Positive and negative attraction intensity of cities in northern Xinjiang from 2010 to 2020.

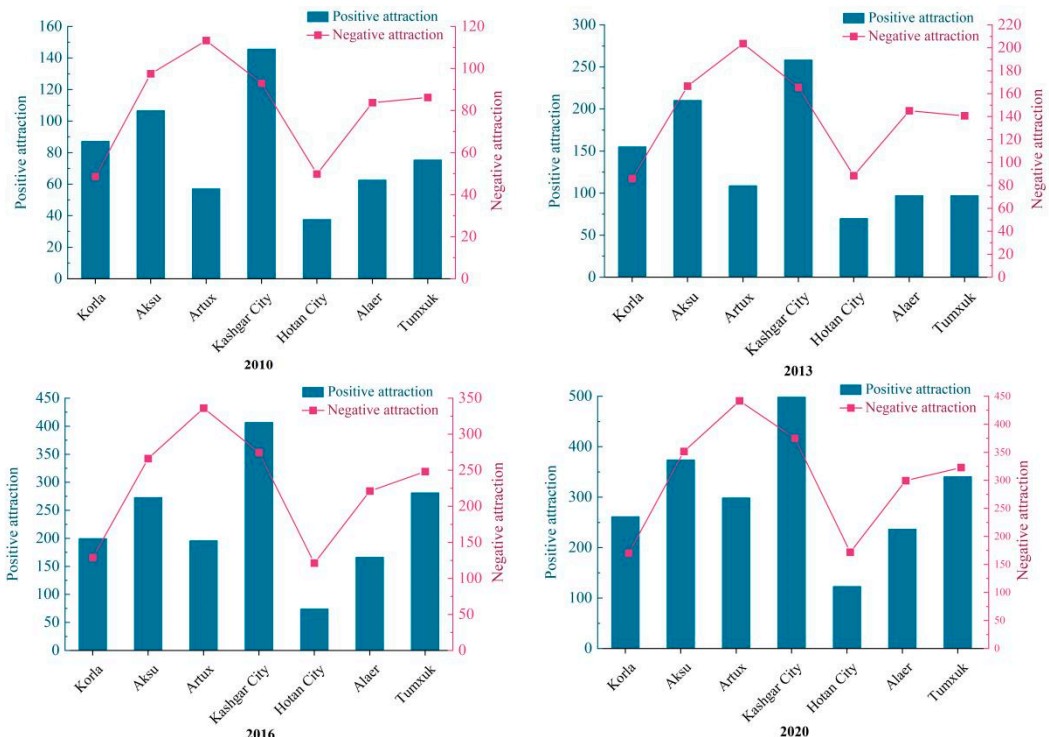

**Figure 4.** Positive and negative attraction intensity of cities in southern Xinjiang from 2010 to 2020.

Urumqi, Karamay, and Shihezi had the top three *p* values in northern Xinjiang in 2010. The top three *p* values were Urumqi, Karamay, and Changji in 2013, 2016, and 2020, respectively. Additionally, throughout the course of the four years, Urumqi's *p* value has outperformed its *n* value, demonstrating a positive attractiveness dominance. In 2010, negative attraction predominated in Karamay, However, positive attraction predominated in 2013 and later. The other cities in northern Xinjiang have a strong deterrent.

Kashgar, Aksu, and Korla had the top three *p* values in the southern Xinjiang region in 2010 and 2013. The top three *p* values shifted to Kashgar, Tumshuk, and Aksu in 2016 and 2020. Additionally, Tumshuk displayed a *p* value and *n* value in 2010 and 2013, whereas Kashgar, Korla, and Aksu did so every four years. After 2013, the *p* value dramatically grew between 2016 and 2020, showing a *p* value > *n* value with a sharp rise in positive appeal. In the four years, the other cities exhibited predominant unattractiveness.

### 4.3. Characterization of the Spatial and Temporal Linkage Structure of Cities

A combination of outward radiation and outward dependence is represented by the difference between a city's *p* value and its *n* value (*p-n*) or the difference between a city's overall positive attractiveness and its total negative attractiveness. If the difference is positive, it indicates that the city is in the center of the region and, to some extent, promotes the growth of the neighboring cities. If the difference is negative, it indicates that the opposite is true. The city is objectively reflected to be in a peripheral position in the region, more impacted by the core city, or dependent on other cities for development if the difference is negative, which indicates that outward radiation is less than external dependency.

According to Figure 5, the city of Urumqi plays a consistent role as a core radiator in the region, as evidenced by the highest values of (*p-n*) in the northern border region from 2010 to 2020 being all in Urumqi (1579.71, 4348.37) and showing an annual growing trend. The difference between Karamay City in 2010 is negative (−196.45), but the difference changes to become positive in 2013, 2016, and 2020, showing that Karamay City depends on the radiation of the nearby cities from 2010 but changes to primarily radiate outward in 2013. The following cities all have negative (*p-n*) values, indicating that they are on

the periphery of the region and rely on nearby cities for development. In addition, the degree of change in the (*p-n*) values of the cities in the northern border region is decreasing both positively and negatively, indicating that the differences between the cities in the northern border region are gradually decreasing, and the regional integration development is gradually increasing.

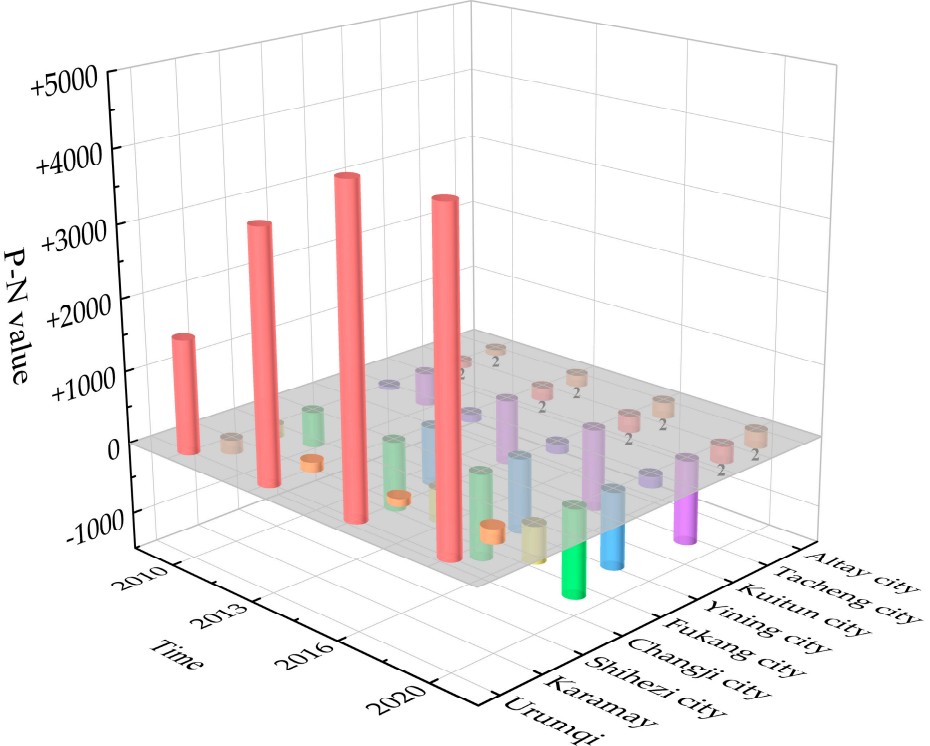

**Figure 5.** Gravity difference of cities in northern Xinjiang from 2010 to 2020.

While Kashgar and Korla are stable Class 1 outward-radiating cities, Artux is a stable Class 3 externally dependent city, and Aksu City was formerly classified as a Class 2 balanced-development city in 2010, 2016, and 2020 before becoming an outward-radiating city in 2013. Hotan and Alaer underwent balanced development in 2010, 2013, and 2016 and external reliance in 2020. Tumxuk was a city with balanced development in 2010, 2013, and 2020; in 2016, it changed into a metropolis that expanded outside.

According to Figure 6, the urban (*p-n*) values in the southern Xinjiang region changed somewhat between 2010 and 2020, with positive (*p-n*) values in Korla, Aksu, and Kashgar. Among them, Korla's (*p-n*) values show a trend of stable increase, demonstrating that the city occupies the region's central position and that the core's radiation influence has grown stronger over time. Kashgar experienced steady growth from 2010 to 2016 before beginning to decline in 2020. This suggests that Kashgar's dominant position in the area has undergone minor changes. Additionally, Tumxuk City's *p-n* values in 2010 and 2013 were negative, while those in 2016 and 2020 were positive. This shows that Tumushuk City has transitioned from a position of passive dependence on other cities for development to one of active development, which also supports the gradual strengthening of its overall strength. From 2010 to 2020, all other cities in the southern Xinjiang region had negative *p-n* values. Based on the aforementioned findings, it can be concluded that over the past ten years, there have been significant changes in the urban gravity structure of the southern Xinjiang region, with multiple cities exhibiting a trend of core effects.

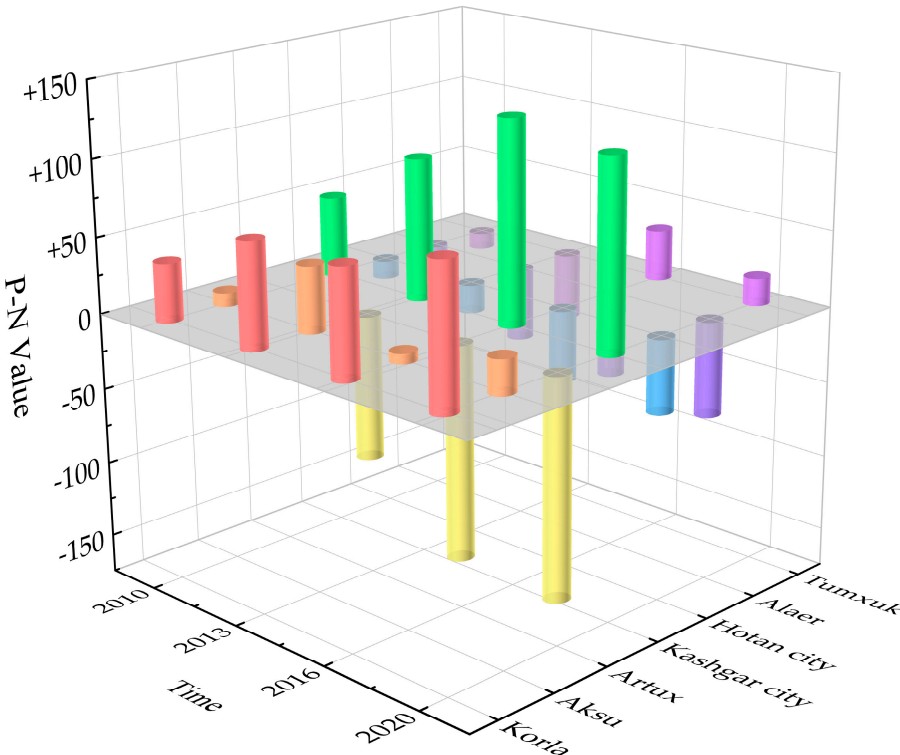

**Figure 6.** Gravity difference of cities in southern Xinjiang from 2010 to 2020.

The *p-n* values of cities in northern and southern Xinjiang were clustered and analyzed year by year, divided into three types of cities: (1) outward radiation type, (2) balanced development type, and (3) outward dependence type [40,41]. This was performed in order to better reflect the development positioning of each city in the urban connection structure (Table 4). The urban connectivity structure's spatiotemporal alterations in northern and southern Xinjiang were depicted (Figure 7). Urumqi, the sole Class 1 city (radiating outwards) in the northern Xinjiang area, has significantly influenced the economic growth of neighboring cities. Due to its superior transportation advantages over other cities, as well as its broad radiation range and potent radiation ability, Urumqi, the capital of Xinjiang, has taken on the role of the region's center city. Two examples of cities with balanced development include Altay, Karamay, Shihezi, Yining, Tacheng, and Yining. Due to their proximity to Urumqi, the "center", and the fact that they are more impacted by core cities, Changji, Fukang, and Kuitun are three examples of externally reliant cities.

**Table 4.** Classification of urban types.

| City Type | City Type Name | City Type Characteristics |
|---|---|---|
| Type 1 | Outward-radiation type | Cities exhibit positive attraction as the dominant factor, radiating and driving the surrounding cities. |
| Type 2 | Balanced-development type | Cities exhibit a balance between positive and negative attraction, with a balance between external radiation and absorption. |
| Type 3 | Outward-dependence type | Cities exhibit a dominant negative attraction and are greatly influenced by surrounding cities, mainly relying on the development of central cities. |

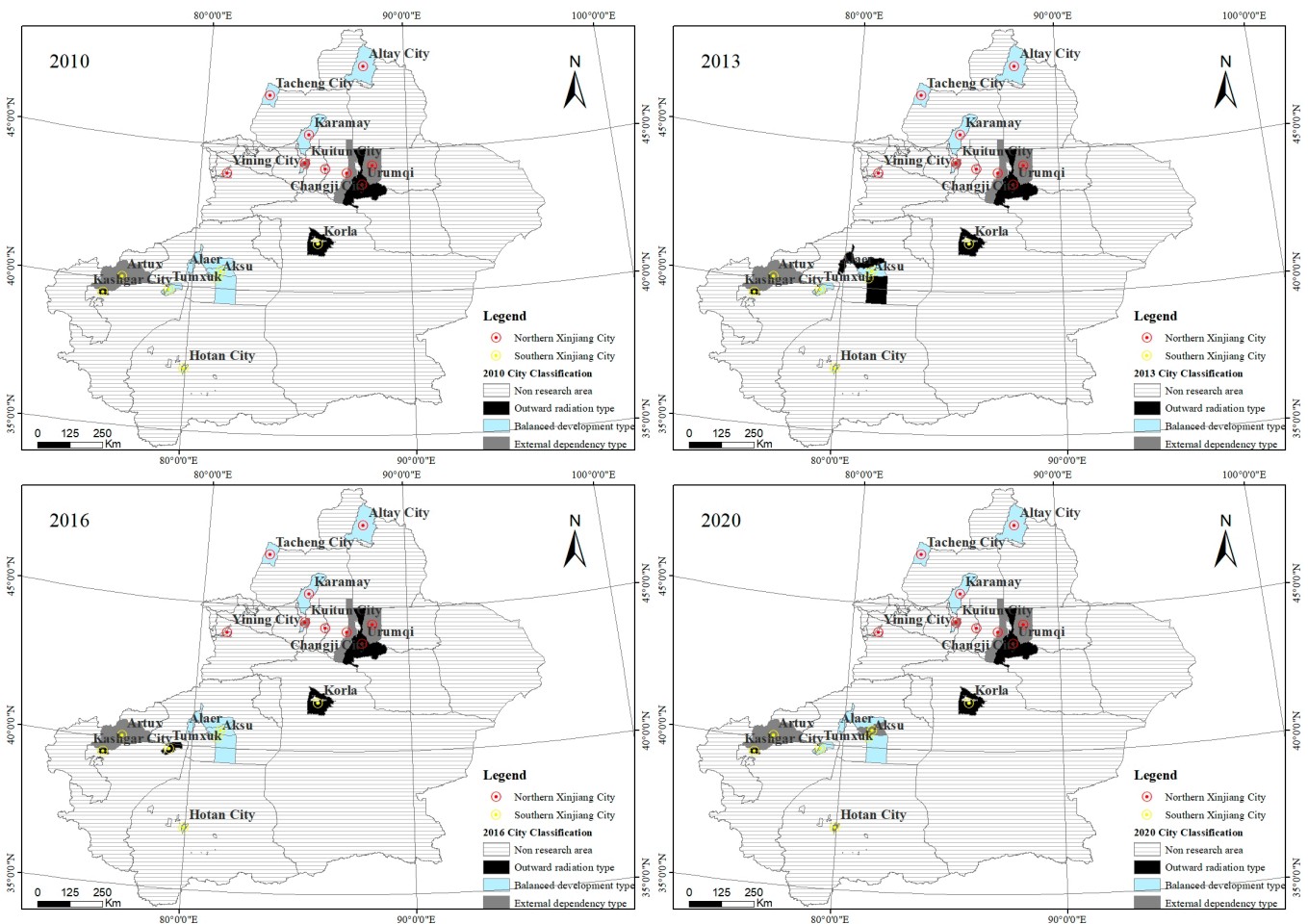

**Figure 7.** Temporal and spatial distribution of urban gravity structure types in northern and southern Xinjiang from 2010 to 2020.

### 4.4. Scope Analysis of Urban Attractiveness

This study calculated the locations of fracture points between cities in northern and southern Xinjiang from 2010 to 2020 using Formula (10) and calculated the magnitude of urban radiation field intensity at the fracture point using Formula (11) (see Tables 4 and 5). This was performed in order to further reflect the trend similarities and differences in the development of urban agglomerations in northern and southern Xinjiang.

**Table 5.** Proportion and field strength of fault points between cities in northern Xinjiang from 2010 to 2020.

| City 1 | City 2 | 2010 | | 2020 | |
|--------|--------|------|------|------|------|
| | | Specific Gravity of Breaking Point | Field Strength | Specific Gravity of Breaking Point | Field Strength |
| Urumqi | Karamay | 0.61 | 0.1407 | 0.59 | 0.2253 |
| Urumqi | Shihezi city | 0.64 | 0.5915 | 0.65 | 0.8907 |
| Urumqi | Changji city | 0.67 | 7.7477 | 0.65 | 12.477 |
| Urumqi | Fukang city | 0.7 | 2.8315 | 0.66 | 4.8344 |
| Urumqi | Yining city | 0.66 | 0.0257 | 0.67 | 0.0375 |
| Urumqi | Kuitun city | 0.66 | 0.1987 | 0.65 | 0.3106 |

**Table 5.** *Cont.*

| City 1 | City 2 | 2010 | | 2020 | |
|---|---|---|---|---|---|
| | | Specific Gravity of Breaking Point | Field Strength | Specific Gravity of Breaking Point | Field Strength |
| Urumqi | Tacheng city | 0.71 | 0.0327 | 0.68 | 0.0537 |
| Urumqi | Altay city | 0.69 | 0.0446 | 0.69 | 0.0676 |
| Karamay city | Shihezi city | 0.54 | 0.1988 | 0.56 | 0.3234 |
| Karamay city | Changji city | 0.56 | 0.0845 | 0.56 | 0.1501 |
| Karamay city | Fukang city | 0.6 | 0.0467 | 0.57 | 0.0898 |
| Karamay city | Yining city | 0.55 | 0.0208 | 0.58 | 0.0325 |
| Karamay city | Kuitun city | 0.56 | 0.3323 | 0.56 | 0.5683 |
| Karamay city | Tacheng city | 0.61 | 0.1002 | 0.6 | 0.1836 |
| Karamay city | Altay city | 0.58 | 0.0371 | 0.6 | 0.0613 |
| Shihenzi city | Changji city | 0.52 | 0.4335 | 0.5 | 0.7123 |
| Shihenzi city | Fukang city | 0.56 | 0.1401 | 0.52 | 0.2507 |
| Shihenzi city | Yining city | 0.51 | 0.0197 | 0.53 | 0.0281 |
| Shihenzi city | Kuitun city | 0.52 | 0.4619 | 0.51 | 0.7276 |
| Shihenzi city | Tacheng city | 0.58 | 0.0246 | 0.54 | 0.0415 |
| Shihenzi city | Altay city | 0.55 | 0.0151 | 0.55 | 0.0228 |
| Changji city | Fukang city | 0.54 | 0.6972 | 0.52 | 1.3887 |
| Changji city | Yining city | 0.49 | 0.0127 | 0.53 | 0.02 |
| Changji city | Kuitun city | 0.49 | 0.1156 | 0.51 | 0.201 |
| Changji city | Tacheng city | 0.55 | 0.0152 | 0.54 | 0.0287 |
| Changji city | Altay city | 0.52 | 0.0211 | 0.55 | 0.0353 |
| Fukang city | Yining city | 0.45 | 0.0089 | 0.51 | 0.0153 |
| Fukang city | Kuitun city | 0.46 | 0.0559 | 0.49 | 0.1064 |
| Fukang city | Tacheng city | 0.51 | 0.01 | 0.53 | 0.0209 |
| Fukang city | Altay city | 0.48 | 0.0181 | 0.53 | 0.0332 |
| Yining city | Kuitun city | 0.51 | 0.028 | 0.48 | 0.0422 |
| Yining city | Tacheng city | 0.56 | 0.0144 | 0.52 | 0.0232 |
| Yining city | Altay city | 0.53 | 0.0051 | 0.52 | 0.0072 |
| Kuitun city | Tacheng city | 0.56 | 0.0312 | 0.54 | 0.0561 |
| Kuitun city | Altay city | 0.53 | 0.0163 | 0.54 | 0.026 |
| Tacheng city | Altay city | 0.47 | 0.0105 | 0.5 | 0.0181 |

4.4.1. Analysis of Urban Breakpoints and Field Strengths

Table 5 shows that the field strength of each city at the fault point in the northern Xinjiang region as a whole increases year after year, with the strongest field strength at the fault point between Urumqi and Changji and the weakest field strength at the fault point between Yining and Altay. In 2010 and 2020, the proportion of fault points ranged from 0.61 to 0.71 and 0.59 to 0.67 in Urumqi, 0.49 to 0.60 and 0.41 to 0.60 in Karamay, 0.36 to 0.58 and 0.34 to 0.55 in Shihezi, 0.33 to 0.55 and 0.35 to 0.55 in Changji, 0.3 to 0.51 and 0.34 to 0.53 in Fukang, 0.34 to 0.56 and 0.33 to 0.52 in Yining, 0.34 to 0.56 and 0.35 to 0.54 in Kuitun, 0.29 to 0.47 and 0.32 to 0.50 in Tacheng, 0.31 to 0.52 and 0.31 to 0.50 in Altay. The Urumqi, Karamay, Shihezi, and Yining fault spots' combined scope is getting smaller every year, which suggests that their radiation effects on the cities in their immediate vicinity are lessening. Contrarily, the Fukang and Tacheng fault spots are growing in size while having less of an impact on the nearby cities' radiation levels.

With the strongest field strength at the fault point between Kashgar and Artux and the weakest field strength at the fault point between Korla and Hotan, as shown in Table 6, the field strength at the fault point in various cities in the southern Xinjiang region also exhibits an increasing trend year after year. Between 2010 and 2020, the specific gravity of the fault points varied between 0.54 and 0.53 at Korla, 0.44 and 0.46 at Aksu, 0.37 and 0.42 at Artux, 0.46 and 0.47 at Kashgar, 0.41 and 0.41 at Hotan, 0.41 and 0.46 at Alaer, and 0.42 and 0.46 at Tumshuk. The Korla and Aksu fault spots' area has been shrinking year over year, which points to a tendency in their radiation influence on neighboring cities that is

waning. On the other hand, the Tumxuk fault point's area has been growing year over year, indicating a tendency in the radiation influence on neighboring cities that is waning. There is no discernible variation in the radiation of each city to the surrounding areas according to the gradual regional stability of the fault point scope in other cities.

**Table 6.** Proportion and field strength of fault points between cities in southern Xinjiang from 2010 to 2020.

| City 1 | City 2 | 2010 | | 2020 | |
|---|---|---|---|---|---|
| | | Specific Gravity of Breaking Point | Field Strength | Specific Gravity of Breaking Point | Field Strength |
| Korla | Aksu | 0.56 | 0.0209 | 0.54 | 0.036 |
| Korla | Artux | 0.63 | 0.0052 | 0.58 | 0.01 |
| Korla | Kashgar city | 0.54 | 0.0067 | 0.53 | 0.0112 |
| Korla | Hotan city | 0.59 | 0.0061 | 0.59 | 0.0102 |
| Korla | Alaer | 0.59 | 0.0188 | 0.57 | 0.0331 |
| Korla | Tumxuk | 0.58 | 0.0099 | 0.54 | 0.0185 |
| Aksu | Artux | 0.58 | 0.0212 | 0.54 | 0.0438 |
| Aksu | Kashgar city | 0.48 | 0.0255 | 0.49 | 0.0453 |
| Aksu | Hotan city | 0.54 | 0.0138 | 0.54 | 0.0248 |
| Aksu | Alaer | 0.54 | 0.2872 | 0.53 | 0.5425 |
| Aksu | Tumxuk | 0.52 | 0.0945 | 0.5 | 0.1896 |
| Artux | Kashgar city | 0.4 | 2.1001 | 0.44 | 4.1759 |
| Artux | Hotan city | 0.46 | 0.0105 | 0.5 | 0.0215 |
| Artux | Alaer | 0.46 | 0.0112 | 0.49 | 0.0242 |
| Artux | Tumxuk | 0.45 | 0.0399 | 0.46 | 0.0914 |
| Kashgar city | Hotan city | 0.56 | 0.0193 | 0.56 | 0.0337 |
| Kashgar city | Alaer | 0.56 | 0.0145 | 0.54 | 0.0264 |
| Kashgar city | Tumxuk | 0.54 | 0.0455 | 0.51 | 0.088 |
| Hotan | Alaer | 0.5 | 0.0184 | 0.49 | 0.0342 |
| Hotan | Tumxuk | 0.48 | 0.0111 | 0.46 | 0.0221 |
| Alaer | Tumxuk | 0.49 | 0.0622 | 0.47 | 0.1292 |

4.4.2. Urban Radius

Overall, the boundary field strength of the strong radiation circle in the northern Xinjiang region is determined by the field strength at the fracture point between Urumqi and Changji, while in the southern Xinjiang region, the boundary field strength is determined by the field strength at the fracture point between Kashgar and Atush. There are principally two causes: first off, the region's two largest cities, Urumqi and Kashgar, have substantial radiation capacities and comparatively powerful fields at their historical breakpoints; second, based on the maximum permitted speed of the motorways, the real traffic distances between the cities are 38.9 km and 44.3 km, respectively. This allows for considerable economic radiation between the cities.

Since all cities, with the exception of the central city, have similar radiation intensities, which can reflect how the city affects its surroundings, the average field strength at the fracture point between other cities and nearby cities was selected as the weak radiation circle's boundary field strength. Due to the fact that the actual distance between the northern Xinjiang region cities of Altay Ili, Korla Kashgar, Korla Hotan, and Korla Artux is over 1000 km, the time difference exceeds 10 h, the transportation distance is too great, and the economic radiation is difficult to access, it is excluded from the calculation of the average field strength because it lacks statistical significance. Figure 7 indicates within- and between-group differences in farmers' livelihood capital before and after land transfer in different modes.

As seen in Figures 8 and 9, this study is based on the spatial comparison of strong and weak radiation radii as well as the trend of changes in strong and weak radiation radii in various cities between 2010 and 2020. The findings reveal the following:

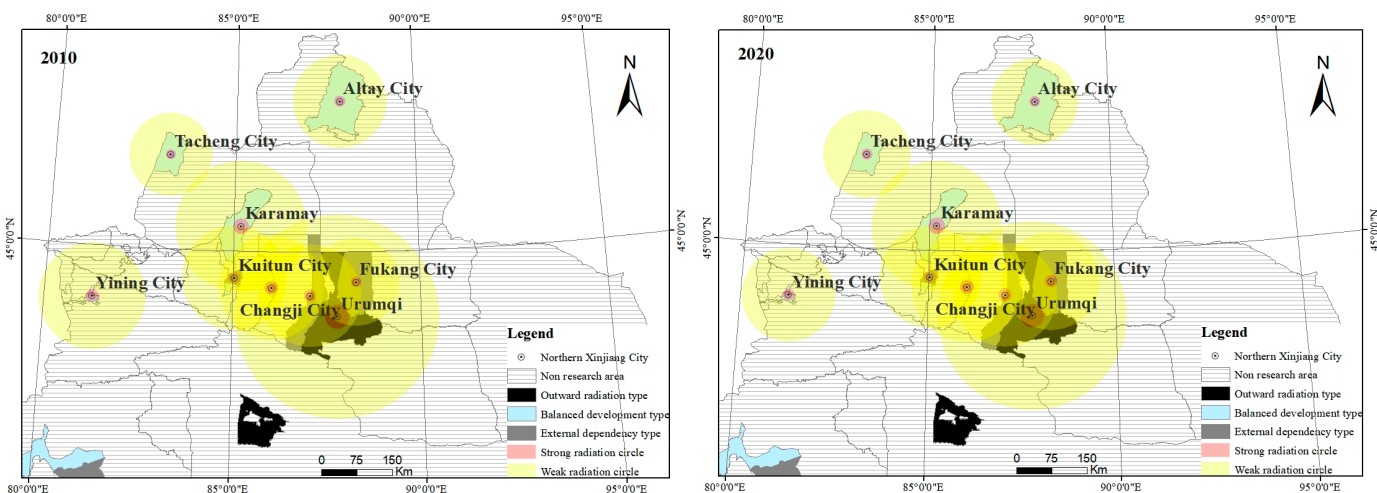

**Figure 8.** Strong and weak radiation range of cities in northern Xinjiang from 2010 to 2020.

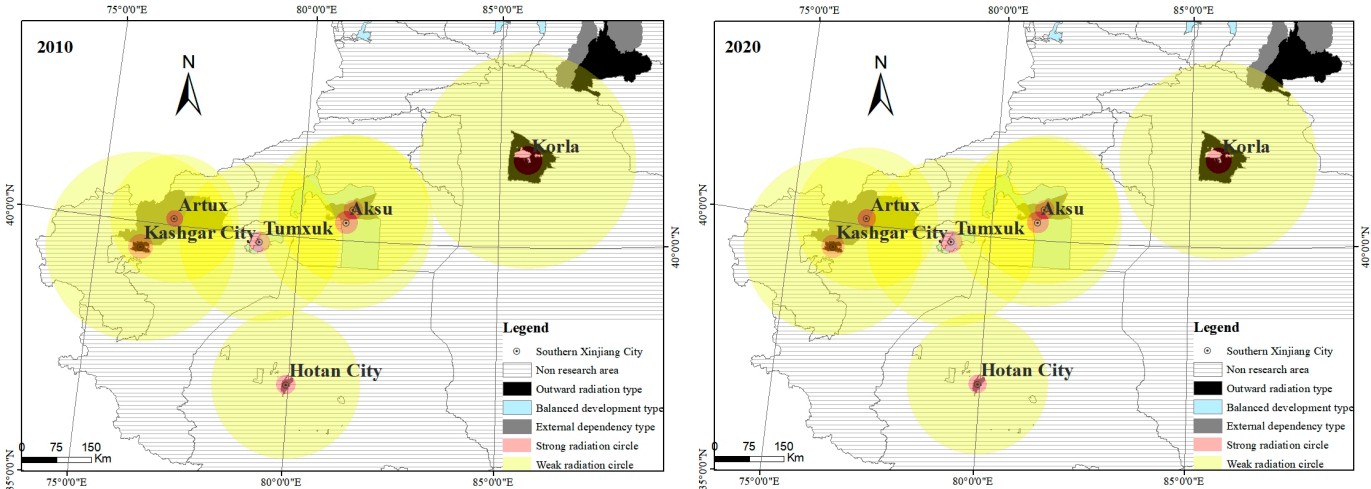

**Figure 9.** Strong and weak radiation range of cities in southern Xinjiang from 2010 to 2020.

The strong and weak radiation radii of Urumqi, Shihezi, Yining, and Altay in the northern Xinjiang region have all decreased, indicating that the surrounding cities have increased their radiation capacity. The expansion of the strong radiation radius of Karamay, Changji, and Kuitun shows that the capacity of these cities is gradually improving, the radiation capacity is gradually increasing, the strong radiation radius of Yining and Fukang has no obvious change, and the weak radiation radius is gradually expanding, which shows that the radiation scope of surrounding cities is expanding and the driving role of surrounding cities is becoming more and more significant. In conclusion, it was discovered that the radiation capacity of the cities of Yining, Changji, Karamay, and Kuitun has gradually increased while the position of Urumqi as the key city in northern Xinjiang has started to wane. Multiple core cities are emerging, and the cross-area of radiation regions of other cities has gradually grown, showing that the trend of regional integration development in northern Xinjiang is becoming more and more important.

The strong and weak radiation radii of Artux and Tumxuk have all shown an expanding trend, indicating that the radiation capacity of Artux and Tumxuk cities is gradually rising and the radiation range to neighboring cities is gradually expanding. These cities are located in the southern Xinjiang region, where the strong and weak radiation radii of Korla, Kashgar, Aksu, Hotan, and Alaer have all shown a decreasing trend. According to the aforementioned data, Kashgar and Korla's central position in the southern Xinjiang region has started to wane, while Tumxuk and Atush's development has started to pick

up. This suggests that the cities in the southern Xinjiang region are beginning to exhibit an integrated pattern.

## 5. Discussion

In the Xinjiang region, urban growth and construction are still in the planning and construction stages. In this study, urban strength, attractiveness, and link strength are calculated and analyzed to reveal the factors that influence how cities in the urban agglomeration of northern and southern Xinjiang develop in tandem. The distinction between the urban agglomerations in northern and southern Xinjiang's different stages of development is reflected in the forms of urban attraction structures. It represents the development tendency of the urban agglomeration of southern and northern Xinjiang through the lens of urban radiation. The findings demonstrate that the strength of urban connections might be limited by the distance between cities. However, to a certain extent, the comprehensive strength of cities can increase the strength of urban spatial connections [42]. Differences primarily distinguish the current development condition of cities in northern and southern Xinjiang. This study analyzes these data and offers the following recommendations:

(1) The northern and southern Xinjiang presents different types of urban agglomeration patterns.

In the southern Xinjiang region, there is a great deal of space between cities, only one major form of transportation, and poor economic growth. Kashgar and Korla have an advantage in terms of the overall strength of the cities from 2010 to 2020. Cities like Aksu and Tumxuk gradually grew as a result of economic growth. Based on the gravitational structures of different cities in southern Xinjiang, it is possible to determine that the urban agglomeration in this region is primarily concentrated around Kashgar and Korla, with Aksu or Tumxuk serving as the secondary center. This region has a "2 + 1 + n" urban agglomeration pattern, with two central cities, one sub-central city, and numerous peripheral cities.

(2) There are individual special types of cities in both northern and southern Xinjiang.

In the study, it was discovered that the northern Xinjiang region's Urumqi city and Kashgar city have significant advantages in the area and have attracted a lot of interest from neighboring cities. Since they are "leading cities" for regional growth, these cities can be considered. We classify Yining city in the northern Xinjiang region and Korla city in the southern Xinjiang region as "island cities" because of their strong comprehensive strengths but remote locations relative to other cities in the area and lack of urban attraction. The "Xinjiang New Urbanization Plan (2021–2035)" passed at the conference in 2021 proposed to accelerate the construction of the urban agglomeration on the northern slope of the Tianshan Mountains and focus on creating a "one circle, one belt, and one group" urban development pattern [43]. Out of these, the northern Xinjiang region should center on Urumqi, establish the Urumqi metropolitan area, propel the growth of neighboring cities like Changji and Fukang, and use Yining city as the regional center to establish the northern Xinjiang urban belt, linking cities like Kuitun, Tacheng, Bole, and Altay for coordinated, linked, and integrated development; the southern Xinjiang region should center on Kashgar, establish a southern Xinjiang urban agglomeration, radiating and propelling the development of southern Xinjiang cities like Korla, Tiemenguan, and Tumushuke.

(3) Suggestions based on urban development in different regions.

This study believes that the urban agglomeration in northern Xinjiang can strengthen the cluster development of small cities and strengthen the agglomeration role of large cities [44]. For instance, the creation of a multi-center driving urban development model with Urumqi serving as the stability center and Kuitun, Karamay, and Yining serving as the sub-centers will enhance communication and cooperation between Urumqi, Kuitun, Karamay, and Yining, forming a "golden triangle" of urban development in northern Xinjiang. Among them, Kuitun has a favorable geographic location as it is situated in the central portion of the region The economic growth of Kuitun should be strengthened, and it is important to encourage the steady growth of the urban small circle into the urban

large circle. The southern Xinjiang region is more fragmented than the northern Xinjiang region, with fewer gravitational crossings and poorer linkages between cities. In order to reduce the "distance" between cities, prioritize the growth of small cities, and increase the overall strength of cities, it is required to upgrade urban transportation infrastructure in the southern Xinjiang region. Korla's unique geographic position can also be used to realize the urban development model connecting northern and southern Xinjiang.

Furthermore, although this study still has several shortcomings, it is a crucial avenue for future investigation:

(1) Methodological limitations. There is still need for improvement in indicator selection, even though this work has enhanced the conventional gravity model and thoroughly assessed urban quality utilizing the urban complete strength evaluation method. For instance, urban gravity and clusters are significantly impacted by the degree of urban economic growth, and the indicators used in the book to indicate this amount of development need to be updated [45,46]. First off, the tourist sector is crucial to Xinjiang's economic growth. As such, the percentage of tourism earnings that goes toward economic development needs to rise. Second, Xinjiang actively encourages the establishment of trade zones and economic development zones as the hub of the Silk Road Economic Belt. Future studies will pay more attention to how changes in logistics, transportation, finance, and other factors affect regional urban connections and urban quality [47,48].

(2) There are restrictions when choosing a city. Xinjiang is split up into local areas and the Xinjiang Production and Construction Corps. At the moment, Xinjiang is pushing hard for the combination of local development and the military [49]. Subsequent studies can emphasize the distinctions in the development models of local and Xinjiang Production and Construction Corps cities, and it can be conducted locally specifically to make recommendations.

## 6. Conclusions

The gravity level, radiation intensity, and radiation range of cities in northern Xinjiang from 2010 to 2020 were investigated in this study using the improved comprehensive gravity model, breakpoint model, and radiation radius model. Nine cities in the northern Xinjiang region and seven cities in the southern Xinjiang region were chosen. The following conclusions were reached:

(1) The quality and all-encompassing urban attraction of cities in northern and southern Xinjiang demonstrated an improving tendency between 2010 and 2020. With various urban agglomeration structures in the north and south of Xinjiang, the quality of cities and the strength of urban links show a characteristic of "high in the north and low in the south".

(2) Between 2010 and 2020, the strength of the geographical connections between the cities in northern and southern Xinjiang gradually grew, with the strength of the connections between the northern and southern cities being noticeably stronger. In northern Xinjiang, the ranking of urban spatial connection intensity is comparatively steady. However, it varies significantly in southern Xinjiang.

(3) Central cities' dominance in both northern and southern Xinjiang is waning. Other cities in the area, including Karamay, Kuitun, and Yining in northern Xinjiang, Tumushuke, and Atushi in southern Xinjiang, are growing in size, their radiation capacity is rising yearly, and their dependence on central cities is waning. Cities in northern and southern Xinjiang are developing progressively in unison and coordination, and regional integration is advancing.

**Author Contributions:** Conceptualization, D.L.; methodology, Y.W. and L.W.; software, Y.M. and D.L.; validation, D.L. and L.X.; formal analysis, H.C.; investigation, Y.M.; resources, Y.W.; data curation, D.L.; writing—original draft preparation, L.W. and D.L.; writing—review and editing, Y.W.; visualization, L.X. and D.L.; supervision, Y.W. and L.W.; project administration, Y.W.; funding acquisition, Y.W. All authors have read and agreed to the published version of the manuscript.

**Funding:** This research was funded by the special project for innovation and development of Shihezi University (CXFZSK202105, CXFZ202217), the Program for Youth Innovation and Cultivation of Talents of Shihezi University (CXPY202223, CXPY202121), and the Science and Technology Plan Project of the Corps (2022DB023).

**Data Availability Statement:** The data presented in this study are available on request from the corresponding author.

**Conflicts of Interest:** The authors declare no conflict of interest.

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
