# Peer review of "Analysis of Spatiotemporal Changes in the Gravitational Structure of Urban Agglomerations in Northern and Southern Xinjiang Based on a Gravitational Model"

_land, doi:10.3390/land13010029_

Round 1

Reviewer 1 Report

Comments and Suggestions for Authors

The paper still has the following shortcomings:

1. The introduction section of the article lacks an overview of relevant research in the research area and does not provide a good reading foundation for readers.

2.Nowadays, there are more diverse modes of transportation, why only choose the time of road travel to measure the distance between cities when improving the model?

3.This chapter believes that the city of Korla has its uniqueness. Can it form a central Xinjiang urban agglomeration centered around Korla?

4.Section 3.3 of the article proposes three types of cities, but lacks descriptive content on classification standards and type characteristics.

5.The discussion section of the article is a relatively important part, which proposes many targeted urban development suggestions. However, the reliability of the suggestions should be further demonstrated in conjunction with existing research.

Comments on the Quality of English Language

none

Author Response

Dear reviewer,

First of all, thank you for your approval of my first round of revision, which gives me great encouragement.

The author carefully answered each question according to the reviewer's requirements, carefully revised the article, and emphasized all modifications to the article.

Due to your suggestions, the revised article will be better and readers will get more valuable information.

Thank you  for your help.

Best regards

Reviewer 2 Report

Comments and Suggestions for Authors

The study is dedicated to the topical issue of urban agglomeration development and spatial connection between cities.

The forecasting of the urban agglomeration development is an important constituent of strategic planning for every country. Complex analysis based on the  spatiotemporal changes allows to find a balanced approach to the effective development of cities and their connection.

The study presents the topical interest. The authors have chosen an approach based on attraction model, fracture point model, and radiation radius model, which allowed to examine the spatial and time changes in the cities of northern Xinjiang.

The following examples: Urban Connection Strength in Northern Xinjiang (Fig. 1) and Gravity Values of Cities in Southern Xinjiang (Fig. 2) are apt.

The study is a complete research, theme of the study is relevant to the Journal research line. Literature review complies with the research.

The study could further benefit, if the authors had presented their proposals on the application of their approach to other regions.

The study could further benefit, if the authors had presented their proposals on the application of their approach to other regions. In addition to that, it would be reasonable to mention, if there were significant changes from 2020 to present. Also, it would be reasonable to address the implementation of the model in case the city types are altered.

The authors should improve the resolution of Fig. 7 "Temporal and Spatial Distribution of Urban Gravity Structure Types in Northern and Southern Xinjiang from 2010 to 2020" for better perception.

Author Response

Dear reviewer,

First of all, thank you for your approval of my first round of revision, which gives me great encouragement.

The author carefully answered each question according to the reviewer's requirements, carefully revised the article, and emphasized all modifications to the article.

Due to your suggestions, the revised article will be better and readers will get more valuable information.

Thank you again for your help.

Best regards

Reviewer 3 Report

Comments and Suggestions for Authors

The data source are not adequately described. Please better describe the construction of the database,

The literature must be carefully expanded.

A statistical and geostatistical analysis of the database is absent.

In my opinion It’s very difficult to interpret the graphs (figures 1 and 2).

Author Response

(The authors gave the same response as above.)

Reviewer 4 Report

Comments and Suggestions for Authors

The paper Based on a gravitational model, analysis of spatiotemporal changes in the gravitational structure of urban agglomerations in northern and southern Xinjiang is elaborated, trying to identify and solve problems related to territorial development starting from the development of cities. For this, the authors choose the gravity model that they modify to obtain results as close as possible to reality.

There are, however, some aspects that are not clear enough or could be improved.

Introduction

Some sentences are not correct: China's urbanization rate will have increased from 19.39% to 63.89% by 2020. It seems to be quoting a work from the past. What is the current rate of urbanization?

Some ideas in the Introduction are redundant and the organization of the text (without paragraphs) makes it difficult to understand.

The national policies in China regarding the promotion of the development of cities and the relations between them for regional development are not specified.

The economic profile of cities plays an important role in development but also in the emergence of clusters but this is cannot be found in the article.

Research Methodology

The authors do not specify how they chose the 11 evaluative factors. On the other hand, the information provided by the factors mentioned above seems redundant (number of residents and population density, to give an example) and a selection methodology (Principal Component Analysis) is necessary.

The methodology has some limitations, which the authors should specify. Depending on the data used, the results of the model that was run may be different and may adequately illustrate reality. For example, if instead of "Share of secondary and tertiary industries" the authors would use data about the main industrial activities of each city, the presence of a rare industrial activity (automotive industry) in one of the cities would non-linearly expand the limits of the attraction of that city.

Author Response

(The authors gave the same response as above.)

Round 2

Reviewer 3 Report

Comments and Suggestions for Authors

Accept in present form